# Offensive language detection in low resource languages: A use case of Persian language

**Marzieh Mozafari** \*, **Khouloud Mnassri** ⓘ *, **Reza Farahbakhsh** ⓘ, **Noel Crespi** ⓘ

Samovar, Télécom SudParis, Institut Polytechnique de Paris, Palaiseau, France

\* marzi.mozafari@gmail.com (MM); khouloud.mnassri@telecom-sudparis.eu (KM)

**Data Availability Statement:** All relevant data are within the manuscript.

**Funding:** The author(s) received no specific funding for this work.

## Abstract

THIS ARTICLE USES WORDS OR LANGUAGE THAT IS CONSIDERED PROFANE, VULGAR, OR OFFENSIVE BY SOME READERS.

Different types of abusive content such as offensive language, hate speech, aggression, etc. have become prevalent in social media and many efforts have been dedicated to automatically detect this phenomenon in different resource-rich languages such as English. This is mainly due to the comparative lack of annotated data related to offensive language in low-resource languages, especially the ones spoken in Asian countries. To reduce the vulnerability among social media users from these regions, it is crucial to address the problem of offensive language in such low-resource languages. Hence, we present a new corpus of Persian offensive language consisting of 6,000 out of 520,000 randomly sampled microblog posts from X (Twitter) to deal with offensive language detection in Persian as a low-resource language in this area. We introduce a method for creating the corpus and annotating it according to the annotation practices of recent efforts for some benchmark datasets in other languages which results in categorizing offensive language and the target of offense as well. We perform extensive experiments with three classifiers in different levels of annotation with a number of classical Machine Learning (ML), Deep learning (DL), and transformer-based neural networks including monolingual and multilingual pre-trained language models. Furthermore, we propose an ensemble model integrating the aforementioned models to boost the performance of our offensive language detection task. Initial results on single models indicate that SVM trained on character or word $n$-grams are the best performing models accompanying monolingual transformer-based pre-trained language model Pars-BERT in identifying offensive vs non-offensive content, targeted vs untargeted offense, and offensive towards individual or group. In addition, the stacking ensemble model outperforms the single models by a substantial margin, obtaining 5% respective macro F1-score improvement for three levels of annotation.

## Introduction

*Disclaimer: This article uses words or language that is considered profane, vulgar or offensive by some readers. Due to the topic studied in this article, quoting offensive language is academically*

**Competing interests:** The authors have declared that no competing interests exist.

*justified but we nor PLOS in no way endorse the use of these words or the content of the quotes. Likewise, the quotes do not represent the opinions of us or that of PLOS, and we condemn online harassment and offensive language.*

The growing ubiquity of user-generated content in social media raises concerns about potential abusive behavior such as threatening or harassing other users, cyberbullying, hate speech, racial and sexual discrimination, etc. for both government organizations and online communities and platforms. Therefore, it is essential to tackle offensive language as one of the most common abusive content invading social media by using automatic abusive language detection systems.

Recently great efforts have been taken to investigate the issue of hate speech detection and offensive language identification for different languages in social media; including various competitions such as Kaggle's Toxic Comment Classification Challenge (https://www.kaggle.com/c/jigsaw-toxic-comment-classification-challenge/), Jigsaw Multilingual Toxic Comment Classification (https://www.kaggle.com/c/jigsaw-multilingual-toxic-comment-classification), and conferences and workshops such as SemEval [1], GermEval [2], HatEval [3], EVALITA hate-speech detection task [4], the first [5], second [6], and third [7] editions of the Workshop on Abusive Language Online (https://sites.google.com/view/alw3/), the Second Workshop on Trolling, Aggression and Cyberbullying (https://sites.google.com/view/trac2/home [34]), etc. Furthermore, a great interest has been evidenced in providing annotated corpora in different aspects of offensive language such as Racism and Sexism [8], Hate and Offensive [9], Hate and NoHate [10], Non-aggressive, Overtly-aggressive or Covertly-aggressive [11], Misogynous and Non-misogynous [4], and Not Offensive and Offensive [12].

Although a major research effort has been dedicated into the investigation of hate speech and offensive language in English [8, 9, 12–16], creating annotated corpora and analyzing hatred and offensive contents in other languages such as Danish [2], Italian [4], Spanish [17], Mexican Spanish [18], Greek [19], Arabic [20, 21], and Turkish [22], or several languages in parallel [23], which have raised many concerns recently.

However, a limited number of previous works have contributed to offensive language detection by exploring the Persian language.

In this paper, we tackle the problem of offensive language detection in Persian language by introducing a three-layered Persian corpus collected from X (Twitter) and annotated by a team of volunteers.

Natural Language Processing (NLP) has been used jointly with classic Machine Learning (ML) [8, 9] and Deep Learning (DL) [13, 15, 24] techniques to propose automated systems with a promising performance for offensive language detection in social media. Recently, transfer learning approaches, as a methodology in which prior knowledge acquired from one task will be applied to solve other related tasks, such as BERT (Bidirectional Encoder Representations from Transformers) [25], XLM (Cross-lingual Language Model Pretraining) [26], and XLM-RoBERTa [27] have achieved promising results in hate speech detection [13, 15] and offensive language identification [28, 29] tasks. In this work, we investigate the usage of monolingual and multilingual pre-trained language models specially ParsBERT (Transformer-based Model for Persian Language Understanding) [30], ALBERT-Persian [31], Multilingual BERT (mBERT) [25], and XLM-Roberta [27] along with different ML and DL models in the performance of identifying offensive language in our Persian corpus, as a low-resource language in this area. We compare different classical ML and DL algorithms with monolingual and multilingual pre-trained language models and report the performance results of the different settings and discuss how different approaches perform in identifying offensive language in three levels of annotation schema.

In addition, to boost the performance of our classification task, we introduce an ensemble stacking model in which we leverage the output probability predictions of single classifiers as

base-level classifiers to train a meta-level classier to identify offensive vs non-offensive, targeted insult vs untargeted offensive content, and targeted offensive towards individual or group more precisely and robustly.

Fig 1 depicts an overall framework of this study at which we address the problem of offensive language in Persian. First, we collect data from X and annotate it according to a three levels annotation schema. After pre-processing step, different classical ML, DL, and transformer-based neural network models will be applied to the annotated corpus to look into the impact of these models in identification of offensive language. Apart from previously studied classical ML and DL models, here, we introduce several transformer-based neural network models for Persian offensive language detection. Finally, to leverage different strengths and weaknesses of the considered models, we combine them in an ensemble model to improve the performance of the offensive language detection task. The datasets created in this study will be made publicly available at https://github.com/marzimzf/Persian_offensive_language_data.

The main contributions in this study are as follows:

1. Building and sharing Persian offensive language corpus along with describing the methodology for collecting tweet data from X and annotation guidelines.

2. Performing comprehensive experiments on annotated Persian corpus to investigate the ability of classical ML, DL, and transformer-based neural network models in addressing Persian offensive language identification task in social media. Furthermore, we focus on transfer learning approach using advanced monolingual and multilingual pre-trained language models such as ParsBERT, ALBERT-Persian, mBERT, and XLM-RoBERTa for Persian offensive language detection task.

3. Introducing a stacked ensemble methodology to improve the performance of the proposed offensive language detection models.

In the next section titled 'Related Work,' we will go through existing studies on offensive language detection techniques, examining both English and other languages, specifically low-resource ones and the Persian language. Following this, in the 'Dataset' section, we will present a comprehensive overview of the creation of our dataset, beginning with the measures taken for data collection, and then, describing the data annotation procedure. Subsequently, in the 'Methodology' section, we will present the methodology of our extensive experiments. This will involve describing the various models we plan to employ, clarifying the experiments executed, and concluding with the presentation of results and our analysis. Finally, we will present our conclusions and summarize future recommendations for further investigation in 'Conclusion' section.

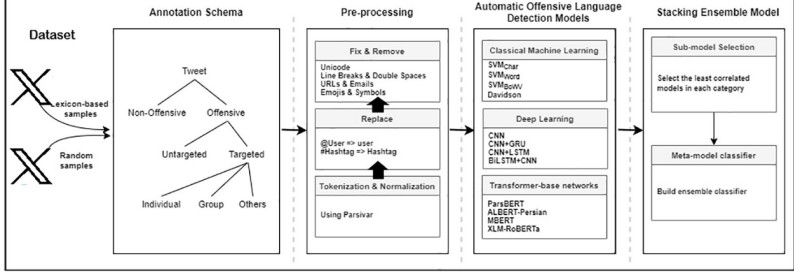

**Fig 1. Workflow of the offensive language detection methodology in Persian language.**

## Related work

Automatic identification of offensive language in online platforms is one of the indubitable necessities of countering online abuse [32]. Over the past decade, there has been increasing interest in leveraging advanced ML and DL techniques for this task [8, 9, 13, 14, 24], mainly focusing on high-resource languages (e.g. English). Here, we discuss a concise overview of different aspects of offensive language detection in social media including definitions and detection techniques, addressing the problem of hate speech in other languages rather than English, and tackling the problem in low-resource languages.

## Offensive language detection techniques

Abusive language is an unwelcome online conduct based on using different remarks intended to be demeaning, humiliating, intimidating, mocking, ridicule, insulting, or belittling. These remarks may or may not be based on an individual's protected status or protected activities such as race, color, religion, sex, national origin, sexual orientation, or gender identity of an individual [8]. By considering abusive language as an umbrella term, that covers different types of online abuse, extensive studies have been done to address hate speech [3, 8–10, 13, 15, 16], offensive language [1, 2, 12], cyberbullying [33, 34], aggression detection [11, 29, 34, 35], and toxicity detection [36].

Employing computational linguistics methods for identification of offensive language and hate speech in social media has been gaining attention in both machine learning [8, 9] and deep learning techniques [37–39]. The most predictive features such as bag of words, word and character $n$-grams, or word embeddings in combination with deep neural networks such as Long Short-Term Memory (LSTM) [37, 39] and Convolutional Neural Network (CNN) [24, 38] are used to address this problem. Furthermore, to train an accurate classifier, different supervised classification algorithms such as Support Vector Machines (SVM) [40], Logistic Regression [8, 9], Naïve Bayes, etc., have been employed.

Recently, transfer learning approach, in which prior knowledge gained from one domain will be applied to solve another problem from another domain, has been more accentuated to identify offensive and hateful content as evidenced in [13, 15, 35, 41]. Promising results yielded by applying different transformer-based language models, e.g., BERT [25] in different NLP tasks indicate the effectiveness of this transfer-learning based approach in different classification tasks as well as offensive language identification. Mozafari et al. [13] proposed different fine-tuning strategies with different neural network architectures as a classifier on top of the pre-trained BERT model to classify tweets as racist, sexist, hate, or offensive. Mnassri et al. [15] introduced an ensemble of multiple fine-tuned BERT models along with various deep neural networks, based on bootstrap aggregating and stacking in hate speech and offensive language detection tasks.

On the other hand, multilingual pre-trained transformer networks, e.g., multilingual BERT [25] and XLM [26] facilitate different downstream NLP tasks, especially in hate speech detection task with low-resource data [42]. A variety of ML and DL models have been proposed to address multilingual offensive language identification in Social Media in SemEval-2020 [1] including word statistics features [43] along with pre-trained transformer networks such as BERT [44], ALBERT and RoBERTa [28], mBERT [45], and XLM-RoBERTa [46]. Considering multilingual hate speech and offensive language detection, Aluru et al. [47] performed an exhaustive experiment on 9 languages from 16 publicly available hatred datasets on Facebook and Twitter. They used LASER and MUSE embeddings to extract sentences and word embeddings, respectively, and leveraged different neural networks models such as CNN-GRU, BERT, and mBERT to identify hateful content in both monolingual and multilingual scenario.

Corazza et al. [48] investigated hate speech detection task in three different languages English, Italian, and German with a combination of neural network architectures (e.g. LSTM, BiLSTM, and GRU) and word-level and tweet-level features (e.g. *n*-grams, word embeddings, emotion lexica, social network-based features, etc.). Ousidhoum et al. [49] introduced a new multilingual dataset comprising English, French, and Arabic tweets from Twitter annotated with a variety of attributes related to Directness, Hostility, Target, Group, and Annotator. They experimented multilingual and multi-task learning approaches to address the problem of hate speech detection on the multilingual dataset. Moreover, Mozafari et al. [23] worked on the identification of hate speech and offensive language, using two diverse collections of publicly available datasets, one for hate speech in 8 languages, and the other for offensive in 6 languages using meta-learning.

## Language-specific abusive language detection

Although many efforts have been dedicated to address the problem of hate speech and offensive language detection in high-resource languages such as English [8, 9, 50], recently concerns have been raised about other languages as well. Emerging recent shared tasks and academic events such as Kaggle's Toxic Comment Classification Challenge in English, Automatic Misogyny Identification (AMI) at IberEval [17] and EVALITA [4] including Spanish and Italian languages respectively, identification of offensive language at GermEval [2, 51] in German language, identification of offensive language at SemEval-2019 [50] for English and SemEval-2020 [1] for Arabic, Danish, English, Greek, and Turkish languages, proceedings of the Workshop on Trolling, Aggression and Cyberbullying Workshops [34, 52], and proceedings of the Workshop on Abusive Language Online [5–7] shows the raising concerns towards hate speech and offensive language detection in different languages. These events and shared tasks mainly focused on different types of this phenomenon such as hate, offensive, misogyny, aggression, etc. in variety of languages. Table 1 summarizes the main concerns of the above events and the datasets and languages that are investigated in these tasks.

Our survey in Table 1 indicates that the attention towards languages with limited resources such as Bangla, Greek, Arabic, Danish, etc. are increasing, and providing annotated data for abusive content in this kind of languages is principal. Many previous studies have been dedicated to studying hate speech and offensive language detection tasks on some specific low-resource languages. In fact, Mubarak et al. [20] provided a list of obscene words and hashtags, which are common patterns in offensive and rude communications, from Twitter along with a large corpus of annotated user comments for obscene and offensive language detection in Arabic language. Guellil et al. [57] investigated the problem of hate speech against politicians in YouTube's comments considering comments written with Arabic, Arabizi, Arabic words written with Latin letters, French, and English. Mubarak et al. [21] proposed a method to build an offensive dataset in Arabic language and analyzed the topics, dialects, and gender mostly associated with offensive content. Pitenis et al. [19] introduced the first Greek annotated dataset for offensive language detection on Twitter, named the Offensive Greek Tweet Dataset (OGTD). Experimenting different ML and DL models on Greek offensive language dataset indicated that LSTM and GRU with attention model results in the best performance. Furthermore, a large corpus from Twitter containing 36232 tweets in Turkish language was created by [22] to address the problem of offensive language in Turkish for the first time. In [18] authors proposed a BERT-based approach along with data augmentation techniques to identify aggressive from non-aggressive tweets written in Mexican Spanish. Considering the automatic detection of hate speech in a code-switching environment, where user writes in one language and then switches to another in the same sentence, authors in [58] proposed a pipeline to extract

**Table 1. Shared tasks in identification of abusive language in different types and languages.**

| Event | Task description | Languages (#sampels) | Platform | year |
|---|---|---|---|---|
| Kaggle's Toxic Comment Classification | **Identification of Different Types of Toxicity** — Threats — Obscenity — Insults — Identity-based hate — | English(300k) | Wikipedia | 2017 |
| AMI at IberEval [17] | **Automatic Misogyny Identification** **Subtask A**—Misogyny Identification: — Misogyny — Non-misogyny — **Subtask B**—Misogynistic Behavior and Target Classification: 1) *Misogyny*: — Dominance — Derailing — Discredit — Stereotype and Objectification — Sexual Harassment and Threat of Violence — 2) *Target*: — Active (individual) — Passive (generic) — | English(3977) Spanish(4138) | Twitter | 2018 |
| AMI at EVALITA [4] | **Automatic Misogyny Identification** **Subtask A**—Misogyny Identification: — Misogyny — Non-misogyny — **Subtask B**—Misogynistic Behavior and Target Classification: 1) *Misogynistic Behavior*: — Dominance — Derailing — Discredit — Stereotype and Objectification — Sexual Harassment and Threat of Violence — 2) *Target Classification*: — Active (individual) — Passive (generic) — | English(5000) Italian(5000) | Twitter | 2018 |
| HaSpeeDe at EVALITA [53] | **Hate Speech Detection on Facebook and Twitter** **Task A**—Hate Speech Detection on Facebook: — Hate — Non-hate — **Task B**—Hate Speech Detection on Twitter: — Hate — Non-hate — **Task C**—Cross-Hate Speech Detection: 1) *Cross-HaSpeeDe-FB*: Train on Facebook and Test on Twitter 2) *Cross-HaSpeeDe-TW*: Train on Twitter and Test on Facebook | Italian: Twitter(4000) Facebook(4000) | Twitter Facebook | 2018 |
| TRAC 2018 [52] | **Aggression Identification** — Overtly Aggressive — Covertly Aggressive — Not — | English(15000) Hindi(15000) | Facebook | 2018 |
| TRAC 2020 [34] | **Aggression Identification**: **Subtask A**—Aggression Identification: — Overtly Aggressive — Covertly Aggressive — Non-aggressive —**Subtask B**—Misogynistic Aggression Identification: — Gendered — Non-gendered — | English(5000) Bangla(5000) Hindi(5000) | YouTube | 2020 |
| GermEval 2018 [51] | **Identification of Offensive Language** **Subtask A**—Coarse-grained Binary Classification: — Offensive — Non-offensive — **Subtask B**—Fine-grained 4-way Classification: — Profanity — Insult — Abuse — Other — | German(8541) | Twitter | 2018 |
| GermEval 2019 [2] | **Identification of Offensive Language** **Subtask A**—Coarse-grained Binary Classification: — Offensive — Non-offensive — **Subtask B**—Fine-grained 4-way Classification: — Profanity — Insult — Abuse — Other — **Subtask C**—Implicit vs. Explicit Classification: — Implicit — Explicit — | German(9915) | Twitter | 2019 |
| HASOC 2019 [54] | **Hate Speech and Offensive Content Identification in Indo-European Languages** **Subtask A**—Hate speech and Offensive language identification: —Hate and Offensive (HOF)—Non Hate-Offensive (NOT)— **Subtask B**—Fine-grained 3-way classification: — Hate speech — Offenive — Profane — **Subtask C**—Type of Offense Classification: — Targeted Insult — Untargeted — | English(8000) German(8000 Code-Mixed Hindi(8000) | Twitter Facebook | 2019 |
| SemEval 2019 (HatEval) [3] | **Multilingual Detection of Hate Speech against Immigrants and Women** **Subtask A**—Hate Speech Detection against Immigrants and Women: — Hateful — Not — **Subtask B**—Aggressive Behavior and Target Classification: 1) *Aggression behavior*: — Aggressive — Non-aggressive — 2) *Target Classification*: — Individual — Generic — | English(13000) Spanish(6600) | Twitter | 2019 |
| SemEval 2019 (OffensEval) [50] | **Identifying and Categorizing Offensive Language** **Subtask A**—Offensive Language Detection: — Offensive — Non-offensive — **Subtask B**—Automatic Categorization of Offensive: — Targeted Insult — Untargeted — **Subtask C**—Offensive Target Identification: — Individual — Group — Other — | English(14100) | Twitter | 2019 |
| SemEval 2020 (OffensEval) [1] | **Multilingual Offensive Language Identification** **Subtask A**—Offensive Language Detection: — Offensive — Non-offensive — **Subtask B**—Automatic Categorization of Offensive: — Targeted Insult — Untargeted — **Subtask C**—Offensive Target Identification: — Individual — Group — Other — | Arabic(10000) Danish(3290) English(14100) Greek(10287) Turkish(35284) + Semi-Supervised OLID English (9089140) | Twitter | 2020 |

*(Continued)*

**Table 1.** (Continued)

| Event | Task description | Languages (#sampels) | Platform | year |
|---|---|---|---|---|
| OSACT4 [55] | **Arabic Offensive Language Detection** <br> **Subtask A**—Offensive Language Detection: — Offensive — Non-offensive — <br> **Subtask B**—Hate Speech Detection: <br> — Hate — Non-hate — | Arabic(10000) | Twitter | 2020 |
| EACL 2021 | **Dravidian Offensive Language Identification** <br> — Not-offensive — offensive-untargeted — offensive-targeted-individual <br> — offensive-targeted-group — offensive-targeted-other — Not-in-indented-language — | Tamil(35139) <br> Malayalam(16010) <br> Kannada(6217) | Youtube | 2021 |
| OSACT5 [56] | **Arabic Fine-Grained Hate Speech Detection** <br> **Subtask A**—Offensive Language Detection: — Offensive — Non-offensive — <br> **Subtask B**—Hate Speech Detection: — Hate — Non-hate — <br> **Subtask C**—Fine-grained type of hate speech detection <br> — Race — Religion — Ideology — Disability — Social Class — Gender — | Arabic($\sim$13K) | Twitter | 2022 |
| RANLP 2023 | **Tamil and Telugu Abusive Comment Detection** <br> **Subtask A**—Offensive Language Detection: — Offensive — Non-offensive — <br> **Subtask B**—Hate Speech Detection: — Hate — Non-hate — <br> **Subtask C**—Fine-grained type of hate speech detection <br> — Race — Religion — Ideology — Disability — Social Class — Gender — | Arabic($\sim$13K) | Twitter | 2022 |

hate speech content in Hindi-English code-switched language (Hinglish) by leveraging profanity modeling, deep graph embeddings, and author profiling.

Low-resource South Asian languages such as Roman Urdu (scripts written in English language characters) and Urdu (scripts written in Urdu language characters) have gained rising attentions [59, 60]. Akhter et al. [59] introduced the first annotated corpus for offensive language detection task in Urdu language and provided profound experiments using ML and DL models to automatically detect abusive comments written in Urdu and Roman Urdu on YouTube's videos. Khan et al. [60] collected and annotated tweets written in Roman Urdu, named Hate Speech Roman Urdu 2020 (HS-RU-20) corpus, in three levels: 1) Neutral or Hostile, 2) Simple or Complex, and 3) Offensive or Hate speech. They applied different ML and DL algorithms including Naive Bayes, Linear Regression, etc. to investigate the effectiveness of supervised learning techniques for hate speech detection in Roman Urdu.

There has been a significant focus on Persian language recently in the domain of offensive language detection. In fact, in 2021, Dehghani et al. [61] assembled a database of 33k Persian abusive language tweets. The database was tested using a list of 648 abusive Persian words. Implementing a deep neural network based on BERT, it gave good performance. Adding to that, Alavi et al. [62] provided a strategy to improve the BERT-based models' performance on English and Persian Offensive Language Detection. They worked on generating more effective word embeddings, altering the 'Attention Mask' input to modify attention probabilities, which led to an improvement of 10% in Persian language. In 2022, we worked on this language along with many other low-resource ones in cross-lingual few-shot hate speech detection [23], we used meta-learning models based on optimization-based and metric-based (MAML and Proto-MAML) techniques. Moreover, Atei et al. [63] introduced Pars-OFF, a three-tiered annotated corpus prepared to identify offensive words in Persian. It is composed of 10K samples. In 2023, Kebriaei et al. [64] employed keyword-based data selection techniques in order to construct a 38K tweet corpus of Persian hateful and offensive language. They used crowdsourcing and an insulting Persian lexicon to collect the data, then they annotated the samples manually. Adding to that, Sheykhlan et al. [65] created the Pars-HAO, a 8k tweets dataset. They used a keyword-based procedure in order to extract samples with higher exposure to hate speech, which were then annotated by three individuals. As baselines, they integrated

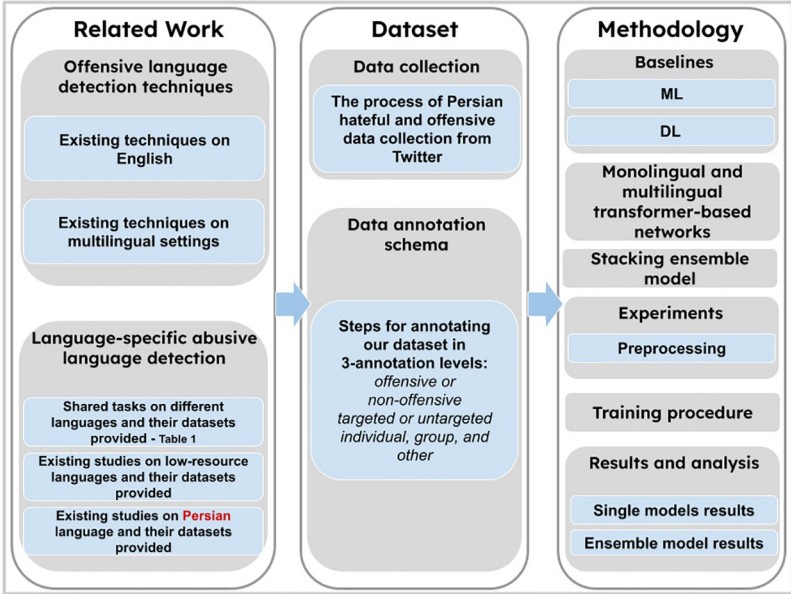

**Fig 2. Paper structure diagram.**

Support Vector Machine (SVM) and Logistic Regression (LR), with the Convolutional Neural Network (CNN) model. Then they used the ensemble Hard Voting to improve the classification task.

Based on those previous approaches, we discovered that offensive language detection in Persian has not been widely addressed in academic research due to the lack of publicly available annotated datasets in the domain, and we believe this study provides interesting insights for this task. The goal of our research is to gather tweet data from X and provide annotation practices in order to build an abusive language corpus in Persian. After that, various comprehensive experiments are carried out to assess how well classical ML, DL, and transformer-based models such as ParsBERT, ALBERT-Persian, mBERT, and XLM-RoBERTa perform in detecting offensive language in Persian. Our paper also presents a stacked ensemble methodology aimed at improving the overall performance of the models.

For a better understanding of the structure of our study, we present a diagram (Fig 2) proposing the flow of our paper. This structure shows the main steps we took during our study from literature review to data creation, models development and training, finishing with results and analysis.

## Dataset

In this section, we explain the method in which the Persian corpus from X is collected and annotated. In addition, we declare our notice regarding the privacy and ethics aspects of users on X as well as GDPR (General Data Protection Regulation) compliance.

### Data collection

We focused on X because it is one of the most widely used microblogging systems and online platforms for sampling offensive and hateful content in different languages [1], and we retrieved Persian tweets from it using Twitter streaming API. We filtered the stream in Persian

language by using both Twister's language identification mechanism (by setting language parameter in the search query as "fa") and some most frequent Persian conjunctions (by setting track parameter in the search query) to prevent crawling samples in other languages similar to Persian such as Urdu. The data was collected using a Python scraper for a two-month interval from June to August 2020. We used two main strategies: (1) random sampling and (2) lexicon-based sampling for data collection, which will be explained in the following.

One of the main difficulties in our data collection process is the fact that Twitter streaming API leads to receiving samples that cover just 1% of all tweets in near to real-time and a very small portion of resulted tweets include offensive or hatred content usually [66]. To investigate the ability of random sampling tweets to reflect offensive language in our data collection process, we selected 400 tweets randomly and inspected them by two experts who are native Persian speakers. Scrutinizing randomly sampled tweets by experts revealed that the actual offensive content constitutes a maximum of 2% selected tweets resulting in an unbalanced and inefficient sampling. Furthermore, the vast majority of offensive samples were related to Iranian political parties and governmental issues at that time or a Persian worldwide trending hashtag: (#Don't_execute, # StopExecutionsInIran), which was launched in support of three young protesters in Iran. Therefore, to prevent a bias against some specific topics or targets during data collection, we used HurtLex seed lexicon [67], to filter more offensive tweets with diversity in topics and targets. HurtLex (https://github.com/valeriobasile/hurtlex) is a multilingual computational lexicon of offensive, aggressive, and hateful words organized in 17 categories in over 50 languages including Persian, with two main labels: conservative and inclusive referring to 'offensive' senses and 'offensive', 'not literally pejorative' and 'negative connotation' senses, respectively. We considered all conservative and inclusive words in 17 categories as keywords to filter tweets in our lexicon-based sampling strategy. Employing random sampling and lexicon-based sampling leaves us with 320$K$ and 200$K$ tweets, respectively. Finally, we selected 3000 tweets randomly from each sampling set (random and lexicon-based) for annotation step.

**The ethical consideration.** Although other information rather than tweet's text such as user demographic statistics, user name, timestamps, location, or social engagement on the platform may result in better understanding of hateful content phenomena, to respect privacy and ethical aspects of users on X as well as GDPR, we did not collect any sensitive and personal information of users. We just collected tweets from public X accounts, eliminating the contact information of users, anonymized and converted all mentions containing @username to a specific and fixed term @user. In the open version of dataset, we are going to publish the annotated corpus in terms of 'TweetID' and 'Label' without the actual text (tweet) and user information.

**Limitations of data collection—Data sampling.** It is important to acknowledge the restrictions associated with the sampling techniques used for data collection. We employed random sampling and lexicon-based sampling techniques to collect tweets related to Persian hate speech and offensive language on Twitter. However, we acknowledge that these procedures may present bias and may not fully capture the perspectives of all our Twitter users. As a result, the nature of our sampling approach may not give the best representation of certain user groups. This limitation could eventually affect the generalizability of our findings. We aim in the future to address these limitations by analyzing alternative sampling methods, such as cluster sampling, to enhance the representativeness of our data and mitigate biases associated with user demographics and access constraints.

## Data annotation schema

Abusive language is an umbrella term that encompasses different types of subtasks such as hate speech, cyberbullying, offensive language, etc. with common or different characteristics

and there is a considerable overlap between these subtasks. To have some kind of uniform understanding of different subtasks related to abusive language and to prevent overlap of their definition and annotation, Waseem et al. [68] unified these subtasks by proposing a 2-fold typology to categorize abusive language into two majority incorporated groups: (1) the target of abuse (an individual or a group) and (2) the nature of the language (explicit or implicit). In addition, Zampieri et al. [12] considered the problem of abusive language definition as a whole and attempted to model the task hierarchically in which the type and the target of offensive content were identified. They proposed a three-layer hierarchical annotation scheme to label the Offensive Language Identification Dataset (OLID), a new English corpus from Twitter, as offensive or not-offensive, its type, and its target. Therefore, Following [12, 68], we developed an annotation protocol for our Persian corpus in three levels as follows:

- **Offensive language detection:** in the first step, tweets are distinguished as *offensive* or *non-offensive*. Similar to [12], tweets having any form of explicit or implicit insults, threats, incitement to hatred and violence, dehumanization, or profane language and swear words are considered as *offensive*. On the other hand, tweets without any form of offense, abuse, or profanity are considered as *non-offensive*.

- **Categorization of offensive language:** After discriminating offensive and non-offensive tweets, we categorize the type of offensive tweets as *targeted* or *untargeted*. Offensive tweets without any specific targeted profanity and swearing are considered untargeted. However, targeted insult refers to any offensive content addressed to an individual, a group, or others.

- **Offensive language target identification:** to make more distinct about the target of offensive contents, similar to [12], we use three target classes: *individual*, *group*, and *other*. If tweets include hateful messages purposely sent to a specific target (e.g. a famous person, a named or unnamed participant in a conversation, etc.), it will be labeled as individual. However, offensive tweets towards many potential receivers as a group of people with the same ethnicity, gender or sexual orientation, political affiliation, religious belief, or other common characteristics are defined as targeted group. Here, we do not consider any crowd of people as a group, but a crowd belongs to a specific unity or individual identity. Therefore, abuse and offense towards some individuals not belonging to our definition of group is considered as individual targeted. We consider another category for tweets in which the target of the offensive language does not belong to individual or group categories, and it is a kind of offense toward an organization, event or issue, situation, etc. as non-human entity target. Using different targets of offensive language in our annotation schema results in different concepts of abusive language. For example, offensive tweets targeted at an individual are known as cyberbullying whereas insults and threats targeted at a group are defined as hate speech.

 **Annotation process.** Since offensive language is a subjective and contextual-based concept that may differ from person to person, culture to culture, or society to society, and Persian is a low-resource language with fewer speakers all over the world in comparison with high-resource languages (e.g. English), employing Persian native speakers for annotating the corpora is crucial. Therefore, we use expert-level annotation approach as a common approach used for annotating low-resource data previously [19, 22, 59] to annotate the Persian corpus. Therefore, three highly educated volunteers from the author's personal contacts, who were Persian native speakers, were enrolled to annotate the corpus. Two annotators were supposed to annotate all the selected tweets at three levels offensive language detection, categorization, and target identification, and in the case of agreement, the final label was set. Otherwise, the

**Table 2. Distribution of annotated data in three levels of annotation schema.** A set of 6,000 out of 520,000 sampled data is randomly selected for annotation process.

| Level-1 | Level-2 | Level-3 | #Samples |
|---|---|---|---|
| Offensive | Targeted | Individual | 702 |
| Offensive | Targeted | Group | 672 |
| Offensive | Targeted | Other | 38 |
| Offensive | Untargeted | - | 212 |
| Non-Offensive | - | - | 4376 |
| **Total** | - | - | 6000 |

third annotator was asked to label the tweet again and then we took a majority vote. Owing to the subjectivity of offensive language identification in three levels of annotation schema and lack of context in tweets, as a short textual data, the annotating process is challenging with low inter-annotator agreement. Annotation consensus for two annotators on three levels of annotation schema was approximately 73%, in which the agreements in the first level of annotation schema (offensive vs non-offensive) was very high as 86%, in the second level 75%, and in the third level 60%. In the case of disagreement, the third annotator judged. The distribution of labeled data in the three levels of annotation is presented in Table 2.

A few examples of annotated instances along with their categories for each level of the annotation schema are presented in Fig 3 in the 'Supporting information' section [22]. We include both Persian and its English-translated versions of tweets for ease of reading.

## Methodology

In this section, we explain in detail different classical machine learning algorithms, deep learning, and transfer learning approaches along with different feature engineering techniques used in this study. We use both classical algorithms of classification (e.g. SVM) and deep learning algorithms (e.g. CNN, bi-directional LSTM, etc.) as our baselines in comparison with making use of monolingual and multilingual pre-trained language models (e.g. ParsBERT, ALBERT-Persian, mBERT, and XLM-RoBERTa) to investigate this problem. Furthermore, we introduce

**Fig 3. Tweet samples (original and translated) from the annotated data with their categories for each level of the annotation schema.**

a new meta-model based on an ensemble learning technique to identify offensive language more precisely.

## Baselines

We conduct several experiments utilizing classical ML algorithms along with a different combination from a pool of features such as Term Frequency-Inverse Document Frequency (TF-IDF) on character and word $n$-grams, Bag of Word (BoW), word embeddings, Part Of Speech (POS) tagging, sentiment scores, etc. to train automatic offensive language detection models.

**Classical machine learning (ML).** Initially, we start with a simple linear SVM classifier, as a well-performed classifier in this task according to the literature [40], trained with different tweet representations as feature embeddings. We use a set of three feature extraction methods TF-IDF on character $n$-grams and word $n$-grams, where $n$-grams are a contiguous sequence of $n$ characters or words, and Bag of Words Vectors (BoWV) over fastText.

**Features**: to extract character $n$-grams, we consider $n = 2$ to $n = 5$. For word $n$-grams we consider $n = 1$ to $n = 2$ and extract word unigrams and bigrams in each tweet and eliminate words with more than 70% of frequency occurrence in all corpus. At the end, using TF-IDF all word and character $n$-grams are normalized. We use a logistic regression with L1 regularization, to reduce the dimensionality of the feature vectors of TF-IDF character and word $n$-grams. Considering the co-occurrences of each word in each document (tweet) in our annotated corpus, we create a document-term matrix and use the pre-trained word embeddings fastText with an embedding dimension of 300 to get initial vector representation of each word in tweet. The fastText is a static word embeddings representation of tweets that is pre-trained on Persian version of Common Crawl and Wikipedia (https://fasttext.cc/docs/en/crawl-vectors.html) using fastText model [69]. The average of fastText vector of words in each tweet is considered as tweet representation.

To investigate the impact of other text-mining features such as sentiment analysis scores, linguistic features, etc. on offensive language detection in Persian, we re-implement a state-of-the-art SVM-based classifier proposed by Davidson et al. [9] and map its feature extraction part in Persian language. Therefore, different features are extracted using Parsivar (https://github.com/ICTRC/Parsivar) Python package [70]. We normalize and tokenize each tweet and calculate: TF-IDF weighted word $n$-grams (unigram, bigram, and trigram); number of characters, words, and syllables in each tweet; number of user mentions, hashtags, retweets, URLs; TF-IDF weighted of POS tag $n$-grams (unigrams, bigrams, and trigrams of POS tags) in which we filter any candidates with a document frequency lower than 5. Using pertimental (https://github.com/pbarjoueian/pertimental) Python package, we also calculate sentiment polarity scores of each tweet as Negative, Positive, and Neutral. Furthermore, readability scores of each tweet are measured using two metrics Flesch-Kincaid Grade Level and Flesch Reading Ease, with common core measures (words and sentences' length) and different weighting factors, to indicate how difficult a tweet in Persian is to understand. To calculate these scores, we consider the number of sentences in each tweet as fixed number one. After reducing the dimensionality of extracted feature vector using a logistic regression with L1 regularization, we apply a Logistic Regression with L2 regularization algorithm to train our classifier.

Thus, we define multiple classifiers named **SVM$_{Character\ n\text{-grams}}$**, **SVM$_{Word\ n\text{-grams}}$**, and **SVM$_{BoWV}$** accompanying **Davidson** algorithm as classical ML approaches. To summarize the baseline classical machine learning models, we added Table 3. It includes parameters, feature extraction techniques, and the specifications of each model.

**Table 3. Baselines ML models.**

| Model name | Feature extraction method | Parameters | Characteristics |
|---|---|---|---|
| SVM$_{\text{Character } n\text{-grams}}$ | TF-IDF on character n-grams | Logistic regression with L1 regularization | n = 2<br>n = 5 |
| SVM$_{\text{Word } n\text{-grams}}$ | TF-IDF on word n-grams | | n = 1<br>n = 2 |
| SVM$_{\text{BoWV}}$ | BoWV over fastText—accompanying Davidson algorithm | | FastText with an embedding dimension of 300<br>Logistic regression with L2 regularization |

**Deep learning (DL).** We employ a static word embeddings (e.g. fastText) representation of tweets to train different DL models combining convolutional neural networks (CNN), recurrent neural networks (GRU), and long short-term memory networks (LSTM).

Following previous studies on different publicly available datasets in this domain, we implement different DL models proposed by [24, 38, 39] on Persian annotated data. Authors in [24], proposed a CNN model trained on different features such as character $n$-grams, word vector embeddings, randomly generated word vectors, and a combination of character $n$-grams and word vectors to study the problem of hate speech identification. Here, we just use the fastText embeddings of words as word feature vectors, based on semantic information, to train a CNN model. The input of the model uses a 1D convolutional layer with 64 filters with a window size of 4, and it is converted into a fixed length vector using a pooling layer. Then, we add a max pooling layer with a pool size of 2 to capture the most important latent semantic features from the input tweets' sequences. We use the Rectified Linear Unit (ReLU) activation function for CNN layers. To provide output in the form of probabilities for each of two classes in our binary classification task, we use a softmax activation function in the output layer. Finally, we compile the model by adjusting three parameters: loss, optimizer, and metrics. A binary-crossentropy loss function is used along with the Adam optimizer to adjust the learning rate throughout the training and the accuracy (as metric).

CNN together with Gated Recurrent Units (GRUs) [38] or LSTM [39] have also been explored as potential solutions in hate speech detection for other languages. Inspiring [38], we create a deep neural network combining convolutional and GRU neural networks. As the embedding layer, we use the pre-trained fastText embeddings to map each word in tweets' sequences into fixed dimensional real vectors. To avoid overfitting, we add a drop-out layer with a rate of 0.2. Then, a 1D convolutional layer with 100 filters with a window size of 4 is added accompanying a ReLU activation function. To reduce the size of each feature map, the amount of parameters, and computations, we add a 1D max pooling layer with a pool size of 4. Then, the extracted features are fed into the GRU layer. Using a global max pooling layer, the highest values in each feature dimension is selected and the output vector is fed into an output layer with a softmax activation function. To train the model and predict probability distribution over two classes, we use the binary cross entropy loss function and the Adam optimizer.

To create a deep neural network combining convolutional and LSTM neural networks, we use the network architecture proposed in [39]. All the layers, structures, and parameters of this model is the same as CNN+GRU model except for GRU layer. Here, in CNN+LSTM model, we add a LSTM layer instead of GRU to the model.

In addition to the aforementioned models, we introduce a model by combining a bi-directional LSTM (BiLSTM) and CNN networks. As the embedding layer, we use the pre-trained fastText embeddings to map each word in tweets' sequences into fixed dimensional real vectors. To avoid overfitting, we add a drop-out layer with a rate of 0.2. A bi-directional LSTM

**Table 4. Baselines DL models.**

| Model name | Feature extraction method | Parameters | Characteristics |
|---|---|---|---|
| CNN | fastText representation | Softmax activation function<br><br>binary-crossentropy loss function Adam optimizer. | 1D convolutional layer with 64 filters and a window size of 4<br><br>pooling layer<br><br>max pooling layer with a pool size of 2<br><br>ReLU activation function |
| CNN+BiLSTM | fastText representation | Softmax activation function<br><br>binary-crossentropy loss function Adam optimizer. | drop-out layer with a rate of 0.2<br><br>bi-LSTM layer with 128 units<br><br>1D convolutional layer with 100 filters with a window size of 2<br><br>Average pooling<br>Max pooling<br><br>dense layer with 64 units<br><br>ReLU activation function |
| CNN+GRU and CNN+LSTM | fastText representation | Softmax activation function<br><br>binary-crossentropy loss function Adam optimizer. | drop-out layer with a rate of 0.2<br><br>1D convolutional layer with 100 filters with a window size of 4<br><br>ReLU activation function<br><br>1D max pooling layer with a pool size of 4<br><br>global max pooling layer |

layer with 128 units followed by a 1D convolutional layer with 100 filters with a window size of 2 is added. The output of CNN layer is average-pooled and max-pooled globally and the results are concatenated. Then, Features encoded by CNN layer are fed into a dense layer with 64 units and the ReLU activation function. The dense layer is followed by the output layer with softmax activation function. The network is compiled with a binary cross entropy loss function and the Adam optimization algorithm.

Thus, we define multiple classifiers named **CNN**, **CNN+GRU**, and **CNN+LSTM** accompanying **BiLSTM+CNN** as DL approaches to identify offensive language. Our main intuition behind using these neural network architectures is to include both local and global contextual features in our offensive language detection problem. The convolution layer (CNN) will extract local and position-invariant features whereas the LSTM layer considers a long range of context dependencies, semantically, rather than local key-phrases. We have added all the deep learning models we used in Table 4, including feature extraction method, parameters and specifications of each model.

### Monolingual and multilingual transformer-based networks

Here we incorporate context into word embeddings (e.g. fastText) using different transformer-based language models (e.g. ParsBERT, ALBERT-Persian, mBERT, and XLM-Ro-BERTa) and fine-tune different pre-trained contextual representations by training them on our offensive language detection task data. Table 5 summarizes the information of different models used in this study, including their configuration, learning parameters, and training corpora.

**Table 5. Description of the transformer-based neural network models used in identification of offensive language in Persian.**

| Name | Provider | Architecture | Method | Configuration | Training corpora |
|---|---|---|---|---|---|
| ParsBERT | Hooshvare Lab | Google's BERT: Transformer-based Monolingual | Masked Language Modeling Next Sentence Prediction | hidden layers: 12 attention heads: 12 hidden sizes: 768 parameters:110M vocabulary: 100K | Persian corpora (14GB): Wikidumps, MirasText, and six manually crawled text data from a various type of websites |
| ALBERT-Persian | Hooshvare Lab | Google's ALBERT: Transformer-based Monolingual | Masked Language Modeling Sentence Ordering Prediction | hidden layers: 12 attention heads: 12 hidden size: 768 parameters: 12M vocabulary: 100K | Persian corpora (14GB): Wikidumps, MirasText, and six manually crawled text data from a various type of websites |
| mBERT | Google | Transformer based Multilingual | Masked Language Modeling Next Sentence Prediction | hidden layer: 12 attention heads: 12 hidden sizes: 768 parameters: 172M vocabulary: 110K | Entire Wikipedia dump: 104 languages |
| XLM-RoBERTa | Facebook AI team | Transformer based Multilingual | Translation Modeling Causal Language Modeling Masked Language Modeling | hidden layer: 12 attention heads: 12 hidden size: 768 parameters: 270M vocabulary: 250K | CommonCrawl data (2.5TB): 100 languages. |

**ParsBERT** [30]: In this approach, we use a monolingual BERT model pre-trained on large corpora from numerous subjects (e.g., scientific, novels, news) with more than 2M documents, crawled from Internet's web pages in Persian language called ParsBERT (https://github.com/hooshvare/parsbert). ParsBERT is a monolingual pre-trained language model based on BERT architecture with the same configurations as BERT$_{base}$ [25] for Masked Language Modeling (MLM) and Next Sentence Prediction (NSP) tasks. Before fine-tuning ParsBERT model in our downstream task, we first format the input sequences in such a way that each sequence is split into tokens, perpended with the classification token [CLS] to the start and appended the [SEP] token to the end. Then, the sequences are padded to the fixed maximum length of input sequences and attention masks are added to them. Here, we set the maximum sequence length to 128. After feeding input data to the pre-trained model, an additional untrained classification layer will be trained for downstream task. We consider the final hidden state corresponding to the classification token ([CLS]) as the aggregate sequence representation for our offensive language detection classification task. Therefore, we fine-tune ParsBERT on the input data split into 90% and 10% as training and validation sets, respectively, by just adding an output layer as a single linear classifier on top of the pre-trained BERT model.

**ALBERT-Persian** [31]: is a monolingual pre-trained language model with A Lite BERT (ALBERT) architecture [71] which is trained on a massive amount of Persian public corpora (Persian Wikidumps, MirasText) and six other manually crawled text data from a various type of Persian websites (BigBang Page scientific, Chetor lifestyle, Eligasht itinerary, Digikala digital magazine, Ted Talks general conversational, Books novels, storybooks, short stories from old to the contemporary era). ALBERT-Persian has significantly fewer parameters than a traditional BERT architecture. We fine-tune ALBERT-Persian by exactly the same way as ParsBERT.

**mBERT** [25]: is a multilingual task-agnostic language representation model with a 12-layer bidirectional transformer trained on Wikipedia pages of 104 languages with a shared word

piece vocabulary. This model is pre-trained in two tasks masked language model and next sentence prediction and can be fine-tuned for text classification in any of 104 languages including Persian. Here we use mBERT to circumvent having to train a monolingual model for Persian language as a low-resource language and fine-tune it using a single linear classifier on top of the model.

**XLM-RoBERTa** [27]: is a transformer-based multilingual masked language model pre-trained on 100 languages, including Persian, using more than two terabytes of filtered CommonCrawl data. To fine-tune XLM-RoBERTa model on our target classification task, we add a linear layer on top of the pooled output, same as previous models.

As shown in Table 5, we use the Base version of ParsBERT, ALBERT-Persian, mBERT, and XLM-RoBERTa pre-trained models and more details regarding fine-tuning the models and hyperparameters used in this study are included in Section Training procedure.

## Stacking ensemble model

Apart from single classifiers in our classical ML, DL, and monolingual and multilingual transformer-based neural network approaches, we use an ensemble learning technique to improve accuracy of the offensive language detection task with a combination of the aforementioned classifiers. Ensemble learning is a technique in which applying multiple learning algorithms and aggregating their decisions somehow results in better predictive performance than using any of constituent learning algorithms alone [72]. A variety of ensemble techniques have been applied in different applications and problems [73], specifically in offensive language detection [29], aggression identification [74], and hate speech detection [40] to achieve better performance to single classification methods.

According to the feature extraction and learning mechanisms, different aforementioned classifiers capture different aspects of offensive language detection task. For instance, classical ML approaches advantage of syntactical and hand-crafted features such as character and word $n$-grams, number of hashtags and mentions, number of exclamation marks, etc. to understand obfuscated and complex words, but they cannot capture contextual or semantic aspects of offensive language in social media content. On the other hand, in offensive language, context is very domain specific and a lack of vector embedding for some words in fastText pre-trained embeddings may affect ML performance while DL models may suffer from generalization due to the lack of enough training data. Furthermore, transformer-based pre-trained language models (e.g. BERT) have a pre-knowledge of a large corpus and can deal with context better even when there is not a large amount of annotated data [14]. Hence, different advantages and drawbacks of different classical ML, DL, and transformer-based pre-trained language models prompt us to make an ensemble classifier out of them to improve the performance of offensive language detection tasks.

Here we use a stacking ensemble technique in which, using a parallel architecture, all classifiers called base-level classifiers are performed independently and their predictions are fed into a meta learner called meta-level classifier to learn how to best combine the predictions from the classifiers. We consider different models in classical ML, DL, and transformer-based neural networks as base-level classifiers and an SVM as meta-level classifier. The stacking ensemble algorithm is summarized in Algorithm 1. First, we extract all features for different classical ML models (line 1) and prepare the input of DL and transformer-based neural networks according to previous subsections (line 2—line 13). Given the labeled dataset, we use a $k$-fold cross-validation approach on the entire data to learn base-level classifiers separately (line 5—line—line 8) and use their out-of-fold predictions as training features for meta-level classifiers (line 9—line 12).

**Algorithm 1** Stacking with *K*-Fold Cross Validation

**Input:** Training data $\mathcal{D} = \{x_i, y_i\}_{i=1}^{m}$, where $x_i \in$ labeled data and $y_i \in$ [0, 1], and $T$ base-level classifiers

**Output:** An ensemble meta-level classifier $H$

1 *Step 1: Extract required features for classical ML and DL algorithms in base-level classifiers*
2 *Step 2: Adopt a cross-validation approach to prepare a training set for meta-level classifier*
3 Randomly split $\mathcal{D}$ into $K$ equal-size subsets: $\mathcal{D} = \{\mathcal{D}_1, \mathcal{D}_2, \ldots, \mathcal{D}_K\}$
4 **for** $k \leftarrow 1$ to $K$ **do**
5 *Step 2.1: Learn base-level classifiers*
6 **for** $t \leftarrow 1$ to $T$ **do**
7 Learn a classifier $h_{kt}$ from $\mathcal{D}\,\mathcal{D}_k$
8 **end**
9 *Step 2.2: Construct a training set for meta-level classifier*
10 **for** $x_i \in \mathcal{D}_k$ **do**
11 Get a record $\{x'_i, y_i\}$, where $x'_i = \{h_{k1}(x_i), h_{k2}(x_i), \ldots, h_{kT}(x_i)\}$
12 **end**
13 **end**
14 *Step 3: Select T/2 of base-level classifiers as $T'$*
15 Select two least correlated base-level classifiers in each model category: Classical ML, DL, and Transformer-based DL
16 *Step 4: Learn a meta-level classifier among with selected base-level classifiers*
17 Learn a new classifier $h'$ based on the newly constructed dataset: $\{x'_i, y_i\}$, where $x'_i$ is from $T'$
18 *Step 5: Re-Learn base-level classifiers*
19 **for** $t \leftarrow 1$ to $T'$ **do**
20 Learn a classifier $h'_t$ based on $\mathcal{D}$
21 **end**
22 **Return** $H(x) = h'(h_1(x), h_2(x), \ldots, h_{T'}(x))$

To give a more visual presentation of the stacking ensemble learning algorithm we used, we added the diagram in Fig 4.

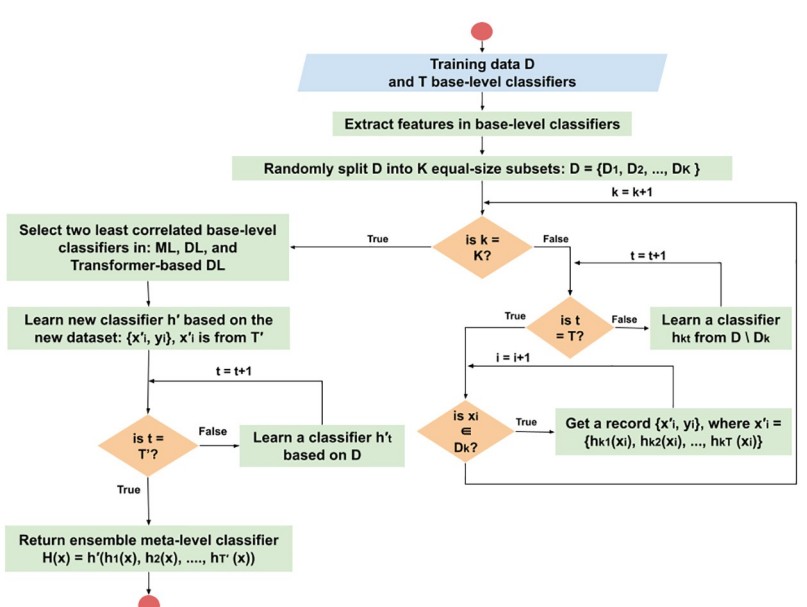

**Fig 4. Diagram of the stacking K-Fold cross validation.**

In order to facilitate further the comprehensibility of the k-fold stacked ensemble learning algorithm we used in our methodology, we provide a general overview of the mathematical presentation of this ensemble learning technique. In fact, as indicated in its algorithm, in stacking with k-fold cross-validation, the procedure involves numerous steps, including training the base-level models on the folds of the data, producing predictions for the validation fold, and then using these predictions as inputs to train the meta-level classifier. The mathematical presentation of this process is as follows, in which we highlighted the corresponding steps of the algorithm:

*1) Base-level models training: (steps 1—2)*
Split the dataset into $K$ folds: $\mathcal{D} = \mathcal{D}_1 \cup \mathcal{D}_2 \cup \ldots \cup \mathcal{D}_K$, where each $\mathcal{D}_k$ illustrates a fold. For each fold $\mathcal{D}_k$:

$$\text{Train } T \text{ base-level classifiers } h_{k,1}, h_{k,2}, \ldots, h_{k,T} \text{ on } \mathcal{D} \; \mathcal{D}_k.$$

*2) Generate Predictions—Construct training set for the meta-level classifier: (step 2.2)*
For each fold $\mathcal{D}_k$:

$$\text{Generate predictions } \hat{Y} = (\hat{x}_i, y_i), \text{ where } \hat{x}_i = h_{k1}(x_i), h_{k2}(x_i), \ldots, h_{kT}(x_i).$$

3) After selection of the least-correlated base-level classifiers, and getting T' base-level classifiers (step 3), we go to:

*4) Stacking: (Steps 4—5)*
Create a new dataset from the predictions we got as input for the meta-level classifier

$$\hat{Y} = [\hat{Y}_1, \hat{Y}_2, \ldots, \hat{Y}_K],$$

where $\hat{Y}_k = [\hat{y}_{k,1}, \hat{y}_{k,2}, \ldots, \hat{y}_{k,T'}]$.

Train a meta-level classifier $H$ using the stacked features $\hat{Y}$, then re-trained it using the main dataset $D$. We finally get $H$ as trained meta-classifier.

## Experiments

This section presents an extensive set of experiments in the identification of offensive language in our Persian corpus comprising classical Machine Learning (ML), Deep Learning (DL), transformer-based models, and the introduced ensemble model. Three binary classifiers are trained for different levels of annotation. The first classifier discriminates *offensive* tweets from *non-offensive*, the second one determines whether an offensive tweet is *targeted* or *untargeted*, and the third one predicts the target of offensive content as *individual* or *group*. We eliminated tweets labeled as *other* due to the sparsity of this class in our dataset (only 38 tweets), and considered only individual and group classes in the third classifier.

**Pre-processing.**   After collecting and annotating our Persian corpus, we perform several text pre-processing steps including: 1) using Parsivar NLP Toolkit https://github.com/ICTRC/Parsivar [70] for normalizing and tokenizing the text; 2) fixing Unicode; 3) removing line breaks, double spaces, emails, URLs, currency symbols, all tweet specific tokens (namely mentions, re-tweet tag, etc.), emoji, punctuation marks, numbers, and non-Persian characters; 4) removing hashtag sign and replacing the hashtag texts by their textual counterparts. In Persian, generally, a hashtag is concatenated with multiple words separated by '_'. Therefore, we split the strings after '#' symbol into their constituent words by removing '_'; 5) correcting the spelling of words using SpellCheck module in Parsivar NLP Toolkit. Overall, Fig 5 provides a clearer visualization of these preprocessing steps.

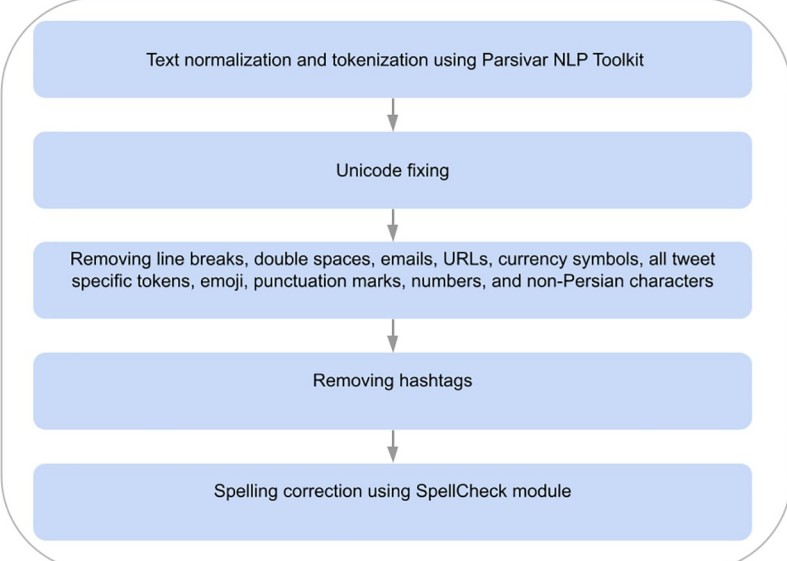

**Fig 5. Preprocessing steps of the dataset.**

We keep stop words in tweets to extract more contextual information from pre-trained language models such as ParsBERT. To fine-tune the transformer-based models, we used specific tokenizer and vocabulary provided by pre-trained models and we did not remove punctuation marks and numbers.

## Training procedure

All classical ML models are performed using scikit-learn (https://github.com/scikit-learn/scikit-learn) python package. The word embeddings dimension in $SVM_{BoWV}$ is fixed to 300. For DL models, we use Keras (https://github.com/keras-team/keras) python package. The initial embedding layer is seeded with a matrix of one embedding in size 300, derived from Persian pre-trained fastText, for each word in the training dataset. All input tweets are padded to sequences of 128 words and in case of a longer or shorter length, truncating or padding with zero values will be applied, respectively. Models are trained for 8 epochs with a batch size of 16. The learning rate of Adam optimizer is set to 1e-5 leading to obtaining more accurate results.

Transformer-based models are fine-tuned employing publicly available transformers (https://github.com/huggingface/transformers) library for Pytorch (namely pytorch-pre-trained-bert). For all considered pre-trained language models (ParsBERT, ALBERT-Persian, mBERT, and XLM-RoBERTa), we utilize the pre-trained model, text tokenizer, and pre-trained WordPiece provided in each pre-trained model to prepare the input sequences for training. The maximum sequence length of the input sentences is set to 128 and in case the input length is shorter or longer, it will be padded with zero values or truncated to the maximum length, respectively. Models are fine-tuned with a batch size of 16 for 3 epochs. An Adam optimizer with a learning rate of 2e-5 is used to minimize the Cross-Entropy loss function. Furthermore, the dropout probability is set to 0.1 for all layers. As offensive language detection is a classification task, we directly modify and fine-tune classification classes of each model (*BertForSequenceClassification* in models with BERT architecture and

*XLMRobertaForSequenceClassification* for XLM-RoBERTa model, in which a linear classification layer is added on top of the pooled output.

As the implementation and execution environment, we use Google Colaboratory tool (https://colab.research.google.com/notebooks/intro.ipynb) with an NVIDIA Tesla K80 GPU and 12G RAM (The code will be released publicly for further research in the camera-ready version of the paper).

**Evaluation metrics:**

We utilize a set of evaluation metrics in order to evaluate the performance of our models. In fact, considering the imbalanced nature of our datasets, we consider using Precision, Recall, F1 score per class, and the macro-averaged F1 score metrics.

Precision (P): measures the accuracy of positive predictions made by the model.

$P = \frac{TP}{TP+FP}$, where *TP* denotes true positives and *FP* denotes false positives.

Recall (R): or sensitivity, measures how much the model can identify all pertinent data samples.

$R = \frac{TP}{TP+FN}$, where *FN* denotes false negatives.

F1-score (F1): the mean of Precision and Recall, providing a balanced examination of the model's performance.

$F1 = 2 \times \frac{P \times R}{P+R}$.

Macro-averaged F1 score: averages the F1 score for each class.

$Macro - F1 = \frac{1}{N} \sum_{i=1}^{N} F_{1i}$, Where *N* is the number of classes, and $F_{1i}$ is the F1-score for class *i*.

## Results and analysis

To evaluate different models in a single or ensemble configuration, we use a *k*-fold cross-validation approach. Due to the imbalanced data that we have, we use Precision (P), Recall (R), and F1-score (F1) per class and macro-averaged F1 score as performance evaluation metrics. For classical ML models, we split data to train and test sets by 0.8 and 0.2. To train DL models and fine-tune other models based on monolingual or multilingual pre-trained language models, we consider 0.1 of the train set as a dev set for hyper-parameter tuning. The reported results are based on the test set.

**Single models results.** Regarding the classical ML, DL, and transformer-based models described in Section Methodology, the first experiment aims to assess and compare the performance of different models along with different features in offensive language detection task in three different levels (*offensive* vs *non-offensive*, *targeted* vs *untargeted*, and *individual* vs *group*). The results of the experiments under *k*-fold cross-validation (*k* = 5) for three classifiers are demonstrated in Tables 6–8 in terms of P, R, and macro F1-score. In all tables, first column indicates the category of trained classifier using classical ML, DL, or transformer-based neural network algorithms. Second and third columns show performance of classifiers per each class in different annotation levels, respectively. Final column indicates the macro-averaged F1 score.

Regarding Table 6, among all models, word *n*-grams are the most discriminator features for identification of offensive content in Persian where SVM classifier trained on word *n*-grams (*n* = 1 to *n* = 2), achieves the best performance 90.8% in terms of macro F1-score. The second best performing model is ParsBERT obtaining 87.8%, following SVM$_{Char}$ and ALBERT-Persian with macro F1-score 87.0% and 84.9%, respectively. Among DL models, CNN+GRU outperforms other models with F1 score 82.7% which confirms the results of previous study [38] for English offensive language detection task. Although SVM$_{Word}$ outperforms other models, a possible reason can be the problem of over-fitting in this traditional classification technique.

**Table 6. Results for offensive language identification (first level).** The bold and underlined numbers represent the first and second best scores, respectively, in each category: classical ML, DL, and transformer-based neural networks.

| Model | | Non-Offensive | | | Offensive | | | |
|---|---|---|---|---|---|---|---|---|
| | | **P** | **R** | **F1** | **P** | **R** | **F1** | **F1 Macro** |
| *Baselines* | | | | | | | | |
| **Classical ML** | SVM$_{Char}$ | 0.944 | 0.971 | 0.958 | 0.844 | 0.730 | 0.783 | 0.870 |
| | SVM$_{Word}$ | 0.995 | 0.930 | 0.961 | 0.759 | 0.979 | 0.855 | **0.908** |
| | SVM$_{BoWV}$ | 0.911 | 0.975 | 0.942 | 0.841 | 0.574 | 0.682 | 0.812 |
| | Davidson [9] | 0.896 | 0.985 | 0.938 | 0.866 | 0.460 | 0.601 | 0.770 |
| **DL** | CNN [24] | 0.909 | 0.969 | 0.938 | 0.807 | 0.567 | 0.666 | 0.802 |
| | CNN + GRU [38] | 0.925 | 0.959 | 0.942 | 0.782 | 0.655 | 0.713 | **0.827** |
| | CNN + LSTM [39] | 0.889 | 0.970 | 0.927 | 0.753 | 0.429 | 0.547 | 0.737 |
| | BiLSTM + CNN | 0.907 | 0.975 | 0.939 | 0.846 | 0.545 | 0.652 | 0.796 |
| *Monolingual/Multilingual language models* | | | | | | | | |
| **Transformer-based DL** | ParsBERT | 0.953 | 0.959 | 0.956 | 0.812 | 0.790 | 0.801 | **0.878** |
| | ALBERT-Persian | 0.930 | 0.971 | 0.950 | 0.840 | 0.675 | 0.749 | 0.849 |
| | mBERT | 0.902 | 0.928 | 0.915 | 0.609 | 0.528 | 0.566 | 0.740 |
| | XLM-RoBERTa | 0.881 | 0.935 | 0.906 | 0.562 | 0.373 | 0.411 | 0.659 |

Comparing the results of different DL and transformer-based models specifies that pre-trained language models such as ParsBERT and ALBERT-Persian that rely on their pre-knowledge existing in their embeddings layers perform better than DL models with fastText embeddings for each word. Furthermore, we can see that all models perform better at identifying non-offensive content compared to offensive where P, R, and F1 score of Non-offensive class are higher than offensive class.

Although the binary classification in the first level of annotation, offensive vs non-offensive, is an important task with a high performance, going deeper into the classification of targeted insult vs untargeted offensive content in the second level of annotation is more challenging.

**Table 7. Results for targeted offensive language identification (second level).** The bold and underlined numbers represent the first and second best scores, respectively, in each category: classical ML, DL, and transformer-based neural networks.

| Model | | Untargeted | | | Targeted | | | |
|---|---|---|---|---|---|---|---|---|
| | | **P** | **R** | **F1** | **P** | **R** | **F1** | **F1 Macro** |
| *Baselines* | | | | | | | | |
| **Classical ML** | SVM$_{Char}$ | 0.760 | 0.904 | 0.826 | 0.983 | 0.951 | 0.967 | **0.896** |
| | SVM$_{Word}$ | 0.645 | 1.000 | 0.784 | 1.000 | 0.912 | 0.953 | 0.869 |
| | SVM$_{BoWV}$ | 0.440 | 0.130 | 0.184 | 0.859 | 0.971 | 0.910 | 0.547 |
| | Davidson [9] | 0.529 | 0.478 | 0.482 | 0.912 | 0.908 | 0.909 | 0.695 |
| **DL** | CNN [24] | 0.573 | 0.115 | 0.180 | 0.859 | 0.979 | 0.914 | 0.547 |
| | CNN + GRU [38] | 0.496 | 0.203 | 0.271 | 0.867 | 0.963 | 0.911 | **0.591** |
| | CNN + LSTM [39] | 0.269 | 0.350 | 0.304 | 0.890 | 0.848 | 0.868 | 0.586 |
| | BiLSTM + CNN | 0.461 | 0.200 | 0.279 | 0.818 | 0.939 | 0.874 | 0.576 |
| *Monolingual/Multilingual language models* | | | | | | | | |
| **Transformer-based DL** | ParsBERT | 0.533 | 0.402 | 0.457 | 0.907 | 0.944 | 0.925 | **0.691** |
| | ALBERT-Persian | 0.347 | 0.186 | 0.227 | 0.868 | 0.974 | 0.917 | 0.572 |
| | mBERT | 0.261 | 0.117 | 0.157 | 0.837 | 0.984 | 0.904 | 0.531 |
| | XLM-RoBERTa | 0.000 | 0.000 | 0.000 | 0.862 | 1.000 | 0.925 | 0.462 |

**Table 8. Results for target type of offensive language identification (third level).** The bold and underlined numbers represent the first and second best scores, respectively, in each category: classical ML, DL, and transformer-based neural networks.

| Model | | Individual | | | Group | | | |
|---|---|---|---|---|---|---|---|---|
| | | P | R | F1 | P | R | F1 | F1 Macro |
| *Baselines* | | | | | | | | |
| **Classical ML** | SVM$_{Char}$ | 0.800 | 0.903 | 0.848 | 0.877 | 0.754 | 0.811 | **0.829** |
| | SVM$_{Word}$ | 0.699 | 0.584 | 0.633 | 0.612 | 0.730 | 0.662 | 0.648 |
| | SVM$_{BoWV}$ | 0.739 | 0.721 | 0.724 | 0.694 | 0.720 | 0.701 | 0.712 |
| | Davidson [9] | 0.887 | 0.983 | 0.933 | 0.849 | 0.427 | 0.568 | <u>0.751</u> |
| **DL** | CNN [25] | 0.711 | 0.711 | 0.702 | 0.677 | 0.678 | 0.667 | <u>0.685</u> |
| | CNN + GRU [38] | 0.784 | 0.772 | 0.778 | 0.722 | 0.735 | 0.728 | **0.753** |
| | CNN + LSTM [39] | 0.657 | 0.707 | 0.681 | 0.612 | 0.555 | 0.582 | 0.632 |
| | BiLSTM + CNN | 0.676 | 0.741 | 0.707 | 0.686 | 0.614 | 0.648 | 0.677 |
| *Monolingual/Multilingual language models* | | | | | | | | |
| **Transformer-based DL** | ParsBERT | 0.753 | 0.833 | 0.787 | 0.786 | 0.702 | 0.736 | **0.772** |
| | ALBERT-Persian | 0.765 | 0.790 | 0.777 | 0.763 | 0.736 | 0.749 | <u>0.763</u> |
| | mBERT | 0.716 | 0.693 | 0.704 | 0.672 | 0.696 | 0.684 | 0.694 |
| | XLM-RoBERTa | 0.521 | 0.891 | 0.654 | 0.293 | 0.108 | 0.106 | 0.374 |

Given Table 7, it is obvious that different classifiers with different features have lower results in identifying whether an offensive tweet is a targeted insult towards an individual or group or it is an untargeted one with general abuse content.

The best macro F1-score, 89.6%, is achieved by training an SVM classifier on character $n$-grams ($n = 2$ to $n = 5$) features. The model trained using word $n$-grams ($n = 1$ to $n = 2$) follows this number by achieving 86.9%. Both ParsBERT and Davidson models provide nearly the same results whereas in DL models there is roughly a 14% reduction (or decrease) in the performance. Among transformer-based models, monolingual pre-trained language models Pars-BERT and ALBERT-Persian outperform multilingual pre-trained language models mBERT and XLM-RoBERTa by achieving macro F1-score 69.1% and 57.2% in comparison with 53.1% and 46.2%, respectively. On the other hand, XLM-RoBERTa as a multilingual pre-trained language model performs the worst among all cases. Although the imbalanced data that we have in the second level of annotation gives rise to decreasing performance among DL and transformer-based models, mBERT is better than XLM-RoBERTa in capturing contextual information in Persian as a low resource language.

The results from Table 8 show that identifying target of offensive content as individual or group in the third level of annotation is more precise than the second level classification results, especially for DL and transformer-based models. SVM classifier trained on character $n$-grams ($n = 2$ to $n = 5$) performs the best and XLM-RoBERTa performs the worst among all models. Again pre-trained language model ParsBERT surpasses other transformer-based and DL models by achieving 77.2% F1-score, where ALBERT-Persian, CNN+GRU, and Davidson follow it by achieving 76.3%, 75.3%, and 75.1%.

Overall we observe that there is no single model outperforming others in the identification of offensive vs non-offensive content, targeted insult vs untargeted offense, and targeted offensive language as individual or group. However, SVM trained on character and word $n$-grams seems to be reliable in most cases where pre-trained language model ParsBERT is the second model with promising results in all three levels of classifications. On the other hand, we believe that the performance of DL models could be improved with a larger amount of labeled data in Persian offensive content, in company with better word embeddings such as fastText

embeddings trained on a specific Persian textual content of social media. Generally, it can be conveyed that it is not easy to distinguish between targeted insult or thread and untargeted offensive language by applying the single models where the performance metrics of Untargeted class, in Table 7, are lower than Targeted class.

**Ensemble model results.** In the second experiment, we investigate the stacked ensemble classifier using a combination of individual classical ML, DL, and transformer-based classifiers. Firstly, we divide input data into a 90:10 split as training and hold-out test sets. Then, we run $k$-fold cross-validation ($k = 5$) on training set to create out-of-fold predictions per each model as base-level classifier. These predictions will be selected and used as training features for meta-level classifier. To create test features for meta-level classifier, we make predictions for the test set (in each fold) and average all 5 predictions per model. The ensemble learning model utilized in this context is outlined in the 'Stacking ensemble model' subsection, where we offer a thorough explanation of our implementation approach for this ensemble algorithm.

As mentioned in Section Methodology, different models capture different characteristics of offensive language and they are skillful in this task in different ways. Therefore, obtaining an appropriate combination of base-level classifiers for ensemble learning is a challenge. As the training data for meta-level classifier is generated from the probability predictions of base-level classifiers' outputs, we consider the correlation between predicted probabilities of each classifier as a linear discriminative metric for base-level model selection.

As depicted in Fig 6, we examine the pairwise Pearson Correlation Coefficient between the predicted probabilities of all base-level classifiers in three levels of annotation. The values range between −1.0 and 1.0 where a value of 1.0 indicates a total positive linear correlation, 0.0 shows a no linear correlation, and −1.0 shows a total negative linear correlation. Here, we consider positive linear correlation or no linear correlation scores for base-level model selection and do not include models with negative linear correlation in generating the ensemble model. From Fig 6a, it is observed that different classical ML, DL, and transformer-based models have different correlations. In classical ML and DL models, there are positive correlations between models' predictions whereas in transformer-based models this value is low except for Pars-BERT and ALBERT-Persian models. This is the same for classifiers in the second-level and third-level of annotation in Fig 6b and 6c except for SVM$_{Word}$ and Davidson models in the third-level. CNN and BiLSTM+CNN models have the highest correlation contrary to SVM$_{BoWV}$ and CNN+GRU models which have the lowest correlation among all first-level classifiers. In the second-level, SVM$_{Char}$ and SVM$_{Word}$ models have the highest correlation where for the third-level CNN and CNN+GRU models are the most correlated.

In this study, we presume that the low correlated or uncorrelated classifiers with high macro F1-score complement each other in an ensemble configuration. Hence, we select two least correlated models in each classical ML, DL, and transformer-based categories as the input of meta-level classifier in stacked ensemble model.

Fig 7 demonstrates the comparison of the ensemble classifier with the individual classifiers selected based on their correlations. It shows the distribution of macro F1 scores in k-fold cross-validation ($k = 5$). Different individual classifiers that are selected as the base-level classifiers for ensemble stacking among with their performance on out-of-fold test set in terms of averaged-macro F1-score are depicted in Fig 7(a)–7(c) for three level of annotation. For more emphasis, we include the average of macro F1 scores from k-fold cross-validation runs and compare the final performance based on that.

For the classification task in the first level of annotation, as shown in Fig 7(a), we can see that the macro F1-score of offensive vs non-offensive language detection task has increased by 5% of its value where the best performing base-level classifier, SVM$_{Word}$, achieved 88.3% while stacking ensemble classifier achieves 93.1%. As shown in Fig 7(b), for the second level

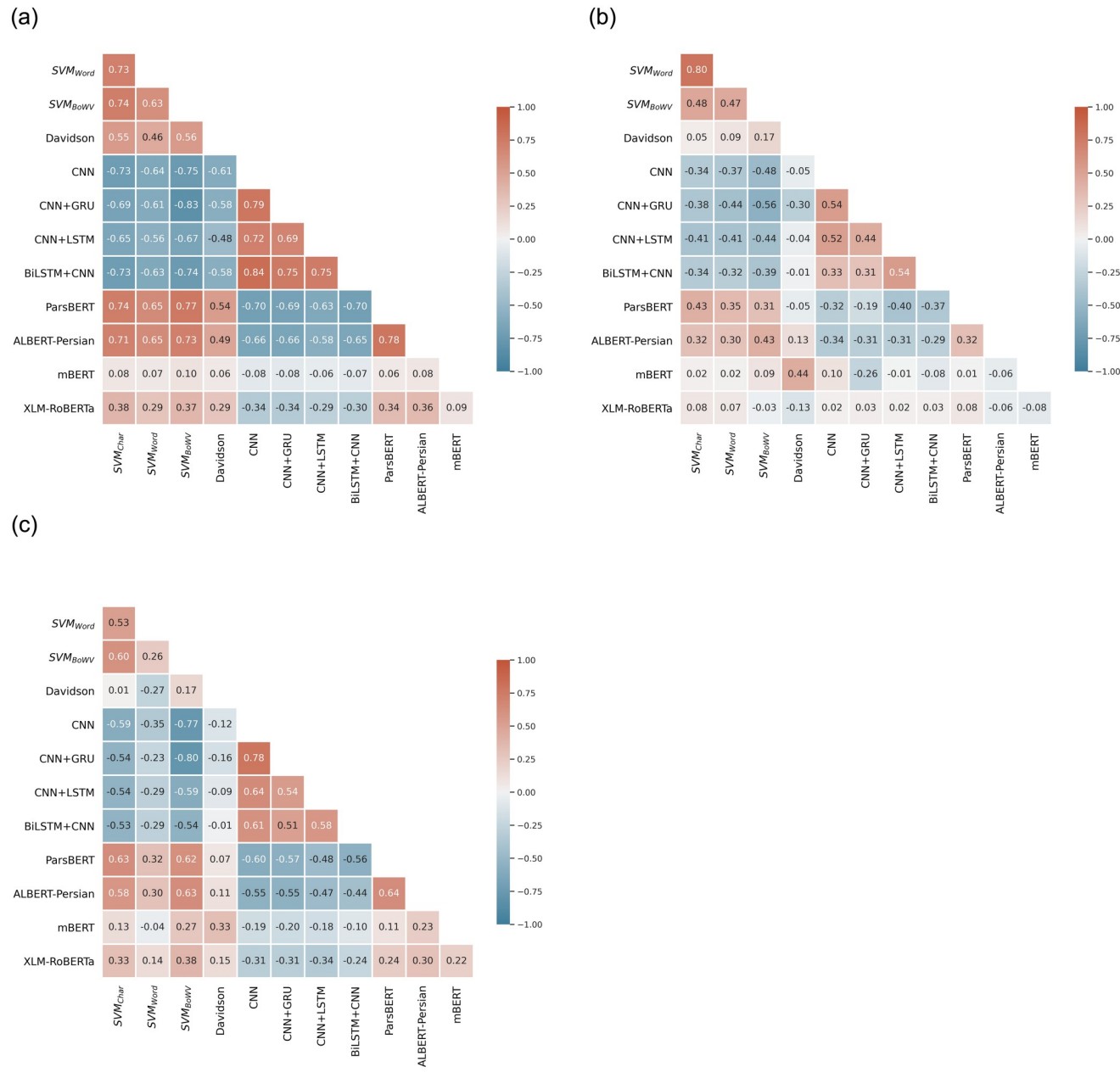

**Fig 6. Pairwise Pearson Correlation Coefficient between the predicted probabilities of different single classifiers on out-of-fold test set.** First level (a) shows the correlation between the output predictions of classifiers trained on offensive vs non-offensive annotated data. Second level (b) shows the correlation between the output predictions of classifiers trained on targeted vs untargeted samples. Third level (c) shows the correlation between the output predictions of classifiers trained on targeted offensive towards individual or group.

classifier, stacking ensemble model outperforms all single base-level classifiers by achieving 90.5% F1 score while on the contrary the best performing base-level classifier, SVM$_{Char}$, achieves 86.2%. As shown in Fig 7(c), ensemble stacking classifier outperforms the best performing single base-level classifier, SVM$_{Char}$ with performance 81.7%, in identifying targeted offensive towards individual or group with 5% of improvement.

In summary it is noticeable that the stacking ensemble method that combines the least correlated classifiers in each category (classical ML, DL, and transformer-based models), with a

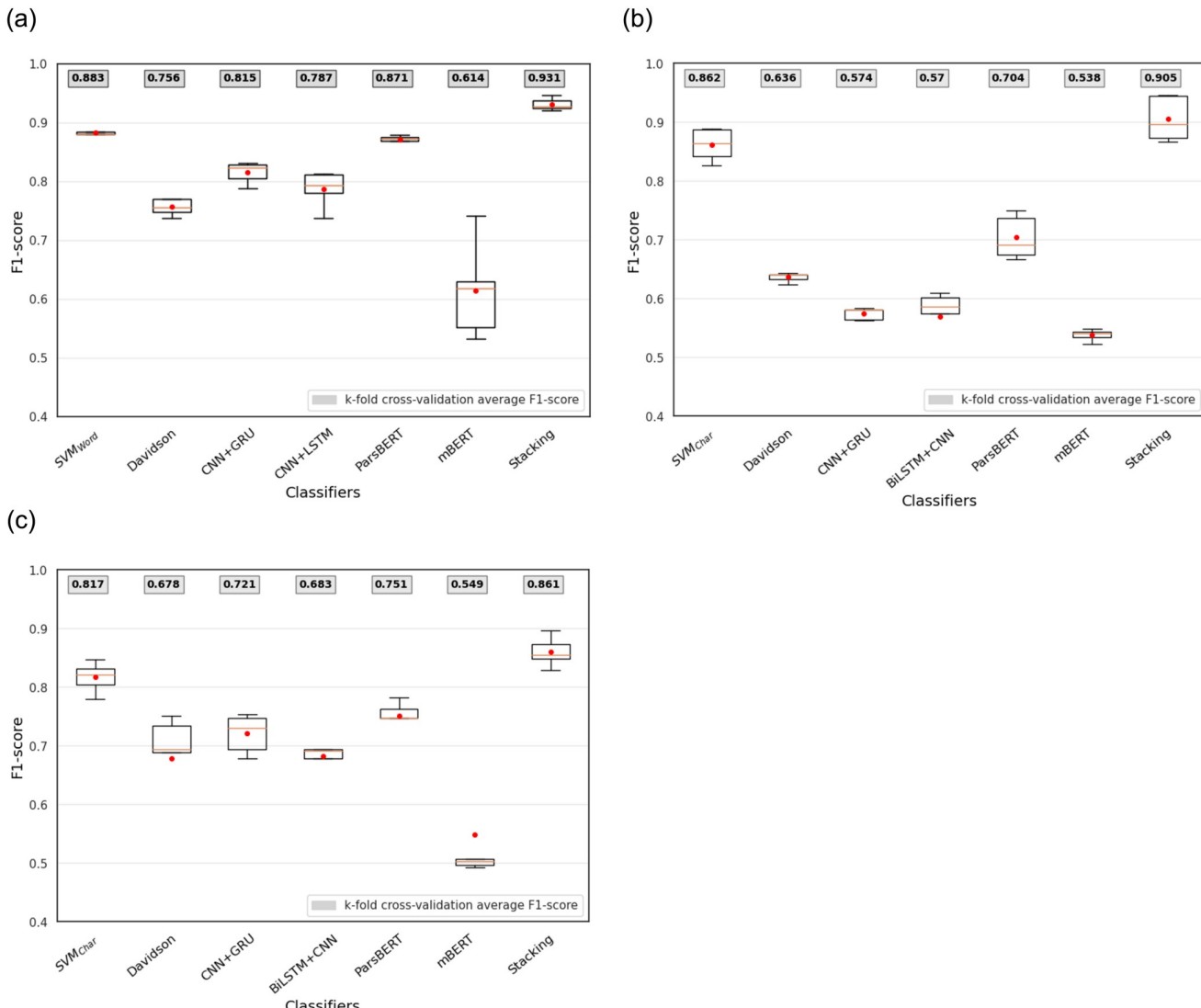

**Fig 7. Offensive language identification performance among all models in three levels of annotation.** First level (a), Second level (b), and Third level (c) indicate performance of selected base-level classifiers accompanying stacking ensemble classifier in identification of offensive vs non-offensive, targeted vs untargeted offensive content, and the target of offensive language towards individual or group, respectively.

variety of knowledge representation and different learning biases, has achieved the highest macro F1-score among the selected classifiers that performed as an individual classifier. Due to the lower noise included in aggregated results of multiple models in comparison with the results of single models, the stacked ensemble classifier has more stability and robustness in its predictions in the identification of offensive language task.

## Conclusion

Automatic detection of offensive language and hate speech on social media for low-resource languages, beyond English, is a rising area of concern among academic researchers with regard to a lack of labeled corpora in such low-resource languages. In this work, we addressed the problem of offensive language detection in Persian as a low-resource language. We collected a

Persian corpus in size of 520,000 from X using both random and lexicon-based sampling techniques and selected 6,000 samples out of it to be annotated with three volunteer native Persian speakers as the first dataset of the Persian language in this task. The corpus was annotated through an existing three-level annotation schema named offensive vs non-offensive, targeted vs untargeted offensive content, and offensive language towards individuals, groups, or others. Afterwards, we conducted several experiments for offensive language detection in Persian language and evaluated the performance of a diverse set of classical ML, DL, and transformer-based neural network models individually. We got outstanding performance results on each of the three levels of annotation, getting macro F1 score of about: 90% using SVM Word n-grams model, 89.6%, and 82% using SVM Character n-grams model, on each of the first, second and third model respectively. Furthermore, we built an ensemble stacking model to increase the performance of the classification task by selecting the least correlated single classifiers with different skills on the problem of offensive language detection. The results signify that among single models, the SVM model trained on character or word *n*-grams followed by pre-trained monolingual model ParsBERT performs the best in the identification of offensive vs non-offensive content, targeted vs untargeted offensive content, and targeted offensive content towards individual or groups in almost all cases. Furthermore, using an ensemble stacking model results in increasing the F1-score of the classification task over single classifiers with a performance improvement of about 5%.

## Future directions

In future work, we aim to deal with our imbalanced dataset by leveraging textual data augmentation techniques for Persian language, using various techniques like generative GPT3 PLM-based models [75] which were used on English data, but we believe we can employ it for low-resource languages like Persian. We aim to train more complex DL and Large Language models even in a multilingual configuration to deal with English-Persian code-mixing offensive language content on online social media. We are also eager to use our dataset to train on the recently released Large Language Models in order to learn more about their efficacy and generalizability, which will improve our comprehension and application of these innovative technologies. Furthermore, we believe that considering linguistic-based characteristics of offensive language in Persian as a low-resource language would give more precise results in detecting such content on social media.

## Author Contributions

**Conceptualization:** Marzieh Mozafari, Khouloud Mnassri, Reza Farahbakhsh, Noel Crespi.

**Formal analysis:** Marzieh Mozafari.

**Investigation:** Marzieh Mozafari, Khouloud Mnassri.

**Methodology:** Marzieh Mozafari.

**Supervision:** Reza Farahbakhsh, Noel Crespi.

**Validation:** Marzieh Mozafari, Khouloud Mnassri, Reza Farahbakhsh, Noel Crespi.

**Visualization:** Marzieh Mozafari, Khouloud Mnassri.

**Writing – original draft:** Marzieh Mozafari, Khouloud Mnassri.

**Writing – review & editing:** Marzieh Mozafari, Khouloud Mnassri, Reza Farahbakhsh, Noel Crespi.

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
