## [Decision Letter · Decision Letter 0]

23 Jan 2024

PONE-D-23-40988Offensive Language Detection in Low Resource Languages: a use case of Persian languagePLOS ONE

Dear Dr. Mnassri,

Thank you for submitting your manuscript to PLOS ONE. After careful consideration, we feel that it has merit but does not fully meet PLOS ONE’s publication criteria as it currently stands. Therefore, we invite you to submit a revised version of the manuscript that addresses the points raised during the review process.

We look forward to receiving your revised manuscript.

Kind regards,

Toqir Rana, Ph.D.

Academic Editor

PLOS ONE

Journal Requirements:

● A clean copy of the edited manuscript (uploaded as the new *manuscript* file).

4. In your Methods section, please include additional information about your dataset and ensure that you have included a statement specifying whether the collection and analysis method complied with the terms and conditions for the source of the data.

5. You indicated that ethical approval was not necessary for your study. We understand that the framework for ethical oversight requirements for studies of this type may differ depending on the setting and we would appreciate some further clarification regarding your research. Could you please provide further details on why your study is exempt from the need for approval and confirmation from your institutional review board or research ethics committee (e.g., in the form of a letter or email correspondence) that ethics review was not necessary for this study? Please include a copy of the correspondence as an "Other" file.

6. Please provide additional details regarding participant consent. In the ethics statement in the Methods and online submission information, please ensure that you have specified (1) whether consent was informed and (2) what type you obtained (for instance, written or verbal, and if verbal, how it was documented and witnessed). If your study included minors, state whether you obtained consent from parents or guardians. If the need for consent was waived by the ethics committee, please include this information.

7. PLOS ONE does not copy edit accepted manuscripts. Therefore, the language in submitted articles must be clear, correct, and unambiguous (http://journals.plos.org/plosone/s/criteria-for-publication#loc-5). We notice that your manuscript contains offensive terms in Fig. 2. We do not think that this text is essential to the main manuscript of your study, and we ask you to move Fig. 2 to a Supporting Information file referenced in the main text.

Please add the following text to the beginning of your Abstract section: "THIS ARTICLE USES WORDS OR LANGUAGE THAT IS CONSIDERED PROFANE, VULGAR, OR OFFENSIVE BY SOME READERS."

Please also add the following text to the beginning of your Introduction section: "This article uses words or language that is considered profane, vulgar or offensive by some readers. Due to the topic studied in this article, quoting offensive language is academically justified but we nor PLOS in no way endorse the use of these words or the content of  the quotes. Likewise, the quotes do not represent the opinions of us or that of PLOS, and we condemn online harassment and offensive language."

We appreciate your attention to these requests.

8. We note that your Data Availability Statement is currently as follows: "All relevant data are within the manuscript and its Supporting Information files."

Reviewers' comments:

Reviewer's Responses to Questions

**Comments to the Author**

1. Is the manuscript technically sound, and do the data support the conclusions?

Reviewer #1: Yes

Reviewer #2: Yes

Reviewer #3: Partly

2. Has the statistical analysis been performed appropriately and rigorously? 

Reviewer #1: Yes

Reviewer #2: Yes

Reviewer #3: N/A

3. Have the authors made all data underlying the findings in their manuscript fully available?

Reviewer #1: Yes

Reviewer #2: No

Reviewer #3: Yes

4. Is the manuscript presented in an intelligible fashion and written in standard English?

Reviewer #1: Yes

Reviewer #2: Yes

Reviewer #3: Yes

5. Review Comments to the Author

Reviewer #1: Contribution: compare many models, present new architecture in hope of solving the problem with improved performance and an experiment. Persian Dataset annotation

Relevant literature: adequately cited and discussed w.r.t different languages and different models for the chosen task. However choices are only cursorily supported by literature

Presentation: good

Clarity: fair enough

Organization: fair enough(most revisions relate to organization of paper)

Reproducible results: yes, model seem adaptable

Purpose of model has been well described/ supported by literature

Input/output and internal processing have been fairly described

Illustrations are well made

Reviewer #2: In Iran do not could access Twitter without restrictions and can access Twitter with VPN.

Persian-speaking Twitter users are mostly outside Iran who have migrated for political reasons, and considering that the vast majority of offensive examples are related to political parties and government issues of Iran, offensive words have been used offensive words by these users.

The second group of Twitter users are Iranian authorities who cannot use offensive words in their tweets.

Other people don't have access to Twitter without VPN and it is not a popular network like Instagram in Iran.

Therefore, the sampling used should be Cluster sampling.

What is the reason for not using other machine learning methods in this manuscript?

Reviewer #3: The manuscript's structure is confusing, making it challenging to comprehend the improvements made in comparison to previous studies. The authors should provide a clear description of their enhancements for better understanding.

The manuscript would benefit from the inclusion of a figure to elucidate the proposed method, enhancing the clarity of the methodology.

The structure of the manuscript appears monotonous. Introducing figures and charts would alleviate this issue, making the content more engaging and accessible to readers.

The author references Ensemble Model Results, but there is a lack of explanation. It is crucial to describe what the Ensemble Model is, providing a clear understanding. Additionally, the inclusion of mathematical equations would be beneficial in elucidating the model's workings.

6. PLOS authors have the option to publish the peer review history of their article (what does this mean?). If published, this will include your full peer review and any attached files.

Reviewer #1: **Yes: **Sadia Tariq

Reviewer #2: No

Reviewer #3: **Yes: **Saman Forouzandeh

---

## [Author Response · Author response to Decision Letter 0]

22 Apr 2024

Replies are included in the rebuttal file.

---

## [Decision Letter · Decision Letter 1]

8 May 2024

Offensive Language Detection in Low Resource Languages: a use case of Persian language

PONE-D-23-40988R1

Dear Dr. Mnassri,

We’re pleased to inform you that your manuscript has been judged scientifically suitable for publication and will be formally accepted for publication once it meets all outstanding technical requirements.

Kind regards,

Toqir Rana, Ph.D.

Academic Editor

PLOS ONE

Additional Editor Comments (optional):

Reviewers' comments:

Reviewer's Responses to Questions

**Comments to the Author**

1. If the authors have adequately addressed your comments raised in a previous round of review and you feel that this manuscript is now acceptable for publication, you may indicate that here to bypass the “Comments to the Author” section, enter your conflict of interest statement in the “Confidential to Editor” section, and submit your "Accept" recommendation.

Reviewer #2: All comments have been addressed

Reviewer #3: All comments have been addressed

2. Is the manuscript technically sound, and do the data support the conclusions?

Reviewer #2: Yes

Reviewer #3: Yes

3. Has the statistical analysis been performed appropriately and rigorously? 

Reviewer #2: Yes

Reviewer #3: Yes

4. Have the authors made all data underlying the findings in their manuscript fully available?

Reviewer #2: Yes

Reviewer #3: Yes

5. Is the manuscript presented in an intelligible fashion and written in standard English?

Reviewer #2: Yes

Reviewer #3: Yes

6. Review Comments to the Author

Reviewer #2: (No Response)

Reviewer #3: The authors responded to my comments, and from my viewpoint, the manuscript has been accepted.

The authors responded to my comments, and from my viewpoint, the manuscript has been accepted.

7. PLOS authors have the option to publish the peer review history of their article (what does this mean?). If published, this will include your full peer review and any attached files.

Reviewer #2: No

Reviewer #3: No

---

## [Editor Report · Acceptance letter]

13 May 2024

PONE-D-23-40988R1 

PLOS ONE

Dear Dr. Mnassri, 

I'm pleased to inform you that your manuscript has been deemed suitable for publication in PLOS ONE. Congratulations! Your manuscript is now being handed over to our production team.

Kind regards, 

on behalf of

Dr. Toqir Rana 

Academic Editor

PLOS ONE